# Reconstruction of a Car–Running Pedestrian Accident Based on a Humanoid Robot Method

**DOI:** 10.3390/s23187882

**Published:** 2023-09-14

**Authors:** Qian Wang, Bo Wei, Zheng Wei, Shang Gao, Xianlong Jin, Peizhong Yang

**Affiliations:** 1School of Mechanical Engineering, Shanghai Jiao Tong University, Shanghai 200240, China; wqhlz1990@sjtu.edu.cn (Q.W.); qinchuang79@163.com (Z.W.);; 2State Key Laboratory of Mechanical System and Vibration, Shanghai Jiao Tong University, Shanghai 200240, China; 3Aerospace System Engineering Shanghai, Shanghai 201109, China; maoli3557@163.com; 4School of Petroleum Engineering, China University of Petroleum, Qingdao 266580, China; gaoshang78567@163.com

**Keywords:** humanoid robot, rigid–flexible coupled human, running pedestrian, traffic accident reconstruction, human biomechanics, car–running pedestrian accident

## Abstract

Due to the characteristics of multibody (MB) and finite element (FE) digital human body models (HBMs), the reconstruction of running pedestrians (RPs) remains a major challenge in traffic accidents (TAs) and new innovative methods are needed. This study presents a novel approach for reconstructing moving pedestrian TAs based on a humanoid robot method to improve the accuracy of analyzing dynamic vehicle–pedestrian collision accidents. Firstly, we applied the theory of humanoid robots to the corresponding joints and centroids of the TNO HBM and implemented the pedestrian running process. Secondly, we used rigid–flexible coupling HBMs to build pedestrians, which can not only simulate running but also analyze human injuries. Then, we validated the feasibility of the RP reconstruction method by comparing the simulated dynamics with the pedestrian in the accident. Next, we extracted the velocity and posture of the pedestrian at the moment of collision and further validated the modeling method through a comparison of human injuries and forensic autopsy results. Finally, by comparing two other cases, we can conclude that there are relative errors in both the pedestrian injury results and the rest position. This comparative analysis is helpful for understanding the differences in injury characteristics between the running pedestrian and the other two cases in TAs.

## 1. Introduction

Car–pedestrian TAs are common collisions, with pedestrian deaths accounting for more than 25% of the annual TA deaths in China [1,2]. The National Highway Traffic Safety Administration (NHTSA) reported that in 2021, pedestrian deaths increased by 12.5% from 2020. Pedestrian fatalities as a percentage of all traffic fatalities increased steadily from 2005 to 2018, from 11.2% to 17.3% [3]. In addition, human injuries caused by TAs are diverse due to the complexity of causes. As well as head and neck injuries, limb and chest injuries are extremely common, especially lower limb fractures and visceral injuries [4].

Digital HBMs are an essential part in TA reconstruction research, and are the most applicable factors in the human–vehicle–road coupled model [5]. MB HBMs are highly efficient and can reconstruct the human movement process, but they cannot simulate biomechanical injuries such as fractures and abrasions. FE HBMs are highly precise and can simulate human biomechanical injuries; they cannot reconstruct the running state of a pedestrian [6,7]. The existing FE (MB) HBMs cannot meet the needs of simultaneously reconstructing the RP collision process and simulating the associated biomechanical injuries [8]. Therefore, to solve this problem, researchers have proposed a hybrid HBM with different functions and characteristics, which has been gradually applied to accident cases [4].

Humanoid robot research has been a hot topic in various fields since its inception. The humanoid running robot achieves the effect of stable walking by applying the theory to the joints and center of mass of the robot [9]. At present, many researchers have achieved good simulations of real walking situations, with high bionicity [10,11]. In addition, humanoid robots have been widely used in various fields [12].

The initial movement state of a pedestrian mainly refers to the posture and velocity immediately before the collision, which are crucial factors affecting the impact kinematics and sustained injuries of the pedestrian when reconstructing a car–pedestrian TA [13,14,15,16]. In addition, several studies have reported that (i) pedestrians are usually moving at impact (either walking or running) [17,18] and (ii) post-impact pedestrian kinematics and injuries are significantly affected by the initial posture and walking velocity [19,20,21,22,23,24]. Furthermore, the US National Highway Traffic Safety Administration (NHTSA) and the Japan Automobile Research Institute (JERI) clearly define the pedestrian dummy postures to be used when simulating pedestrian–vehicle collisions. However, due to the difficulty in obtaining the angles and velocities of body parts and joints at the vehicle–pedestrian collision moment, the initial motion state of pedestrians has rarely been considered in vehicle–pedestrian accident reconstructions, which is ultimately not conducive to traffic police accident identification and forensic injury identification. It further indicates the potential necessity of reconstructing moving pedestrian TAs.

The overall aim of this study is to propose a humanoid-robot-based reconstruction method for vehicle–running pedestrian accidents. First, we applied the principle of a running humanoid robot to the digital HBM to simulate the running and walking processes of pedestrians. Next, we simulated the dynamic process of the RP to reconstruct the car–pedestrian collision accident and to approximate the initial simulation state of the HBM. Then, we compared an accident case with the simulation results to validate the feasibility of this method. Finally, we analyzed the injury and dynamic errors between two other different conditions (ignoring the initial state of the pedestrian); this further illustrated the extent to which the simulation accuracy can be improved by considering the initial motion state of pedestrians.

## 2. Materials and Methods

We used the Facet vehicle model and the rigid–flexible coupled HBM to reconstruct a car–running pedestrian TA based on the humanoid robot method. The dynamic simulation of the running process is based on (i) known accident information and (ii) equations for RP kinematics, thereby reconstructing the RP motion and posture. The customization process is shown in Figure 1.

### 2.1. Facet Vehicle Model

The car model used herein is the Facet model (FM), which is an FE mesh of empty materials. The FM is a three-node or four-node unit and is attached completely to the reference space, rigid body, or deformed body, thereby ensuring accuracy and saving calculation time. The FM was built by importing and exporting the model to and from 3D Max, HYPERMESH, and MADYMO in turn and setting the relevant grids and nodes [25,26,27]. The vehicle FM and the curves of the contact characteristics are shown in Figure 2 and Figure 3, respectively [28]. In MADYMO, the FM realizes the connections between different bodies and the relative motion between different parts; for example, car wheels can be split so that the simulation process is closer to the actual situation.

### 2.2. Rigid–Flexible Coupled HM

Herein, rigid–flexible coupled HBM technology is used to establish a human simulation model for pedestrians, and a fine local FE model is used for the injured body parts. In reconstructing a TA involving an RP, the model is capable of high-precision biomechanical simulation and captures the RP’s kinematics, movement posture, and fracture characteristics. The model combines the advantages of (i) the modeling convenience and high computational efficiency of a multi-rigid-body HBM and (ii) the high bio-fidelity and accurate calculations of an FE HBM [5,26].

The rigid–flexible coupled HBM with scaling modules (Figure 4) is based on the TNO international standard multi-rigid-body HBM in MADYMO [27] and the HUMOS international standard FE HBM [5,26,29]. The body is split into the head, trunk, upper limbs, and lower limbs and then modeled using multi-rigid body and local FE HBMs. The split multi-rigid body and FE body part models are connected by a support method through joints and constraints. The settings related to the joint and constraint forces are in line with those from experiments on human joint forces. The lower limb FE rigid–flexible coupled HBM (Figure 5) was used to study RP collision accidents, and the FE parts comprised three-dimensional models of human tissue such as bones, muscles, and skin. The validation of the HBM can be found in Appendix A.

### 2.3. Determination of Initial Pedestrian Motion State

#### 2.3.1. MADYMO Rigid–Flexible Coupled Dynamics Theory

For pedestrian collision simulations, the initial velocity and initial posture cannot be neglected. The rigid–flexible coupled dynamic simulation method was used for the dynamic running process and the pedestrian–vehicle collision process. We applied the theory of humanoid robots to the rigid–flexible coupled HBM to realize the running process of pedestrians. The simulation process was reconstructed using MADYMO software (https://jp.mathworks.com/products/connections/product_detail/madymo.html, accessed on 11 September 2023).

The governing equation for rigid bodies in the rigid–flexible coupled multi-body system is formulated according to the theory for the dynamics of a multi-rigid body system, the governing equation for a flexible body is formulated using the FE method, and the dynamic equation for the rigid–flexible coupled multi-body system is obtained by combining them. The theory of multi-body dynamics includes Lagrangian and Cartesian mathematical models, and MADYMO adopts Cartesian mathematical models. The dynamic variational equation for rigid bodies moving in space is expressed as
(1)$$\sum_{i=1}^{N}\left(\delta {\dot{r}}_{i}{}^{T}\right.\left.\delta {w^{\prime }}_{i}{}^{T}\right)\left\{-Z_{i}\left(\begin{array}{c} {\ddot{r}}_{i}\\ {\dot{w^{\prime }}}_{i} \end{array}\right)+F_{i}{}^{i}+F_{i}{}^{a}\right\}=0,$$
where $r_{i}$ is the multi-body centroid coordinate matrix, ${w^{\prime }}_{i}$ is the coordinate matrix of the multi-body angular velocity vector in the body local coordinate system, $Z_{i}$ is the multi-body generalized mass array, $F_{i}^{i}$ is the multi-body generalized inertial force array, and $F_{i}^{a}$ is the multi-body generalized active force array.

The dynamic variational equation of the rigid–flexible coupled multi-body system can be deduced from the multi-rigid body dynamic equation as
(2)$$\left\{\begin{array}{l} \sum_{i=1}^{N}\left(\delta {\dot{r}}_{i}^{T}\delta {w^{\prime }}_{i}^{T}\delta {\dot{a}}_{i}^{T}\right)\left\{-Z_{i}\right.\left\{\begin{array}{l} \ddot{r}\\ i\\ {\dot{w}}^{\prime }\\ \ddot{a}\\ i \end{array}\right\}+F_{i}^{i}+F_{i}^{a}+\left.F_{i}^{e}\right\}=0\\ F_{i}^{e}=\left(\begin{array}{c} 0\\ 0\\ -K_{i}a_{i-C_{i}{\dot{a}}_{i}} \end{array}\right) \end{array}\right.,$$
where $a_{i}$ is the flexible body shape function matrix, $K_{i}$ is the flexible body elastic stiffness matrix, $C_{i}$ is the flexible body structural damping matrix, and $F_{i}^{e}$ is the generalized elastic force matrix.

#### 2.3.2. Realization of Pedestrian Running State

The pedestrian posture can be realized by adjusting the joint positions of the HBM. For the FE HBM, which simulates a biomechanical injury of the human body, this can be achieved by applying an initial velocity to the whole translation or a slight mediation of the joints for approximation purposes in simulating pedestrian running or walking. However, this “holistic” method of determining the motion state cannot reflect the intrinsic relationship between the motion state and the walking velocity for each part of the human body. Additionally, the influence of the relative velocity of each part of the human body cannot be considered in the simulation results, especially for the local motion or injuries of the human body, which likely makes the simulation results inaccurate. In addition, human joint rotation angles judged and set based on experience and accident information are often not fully in line with human kinematics.

The pedestrian running process adopts a humanoid robot method based on the concept of a “virtual leg” [11,30]. One end of the leg is set as the center of the body, and the other is set as the origin of the reference coordinate system (Figure 6a). Then, taking the virtual leg length $q_{r}$ and the angle $q_{\theta }$ between the virtual leg and the plumb line as the generalized coordinate system, the differential dynamic equations of the virtual leg motion, whose solution is the motion trajectory in the stage of centroid take off, can be expressed as
(3)$$\left\{\begin{array}{l} mq_{r}^{2}{\ddot{q}}_{\theta }+2mq_{r}{\dot{q}}_{\theta }{\dot{q}}_{r}-mgq_{r}\sin q_{\theta }=0\\ m{\ddot{q}}_{r}-mq_{r}{\dot{q}}_{\theta }^{2}+mg\cos q_{\theta }+k_{r}(q_{r}-q_{r0})=0 \end{array}\right.,$$
where m is the mass of the human body, $q_{r0}$ is the initial length of the spring, and $q_{r}$ is the stiffness coefficient. According to Equation (3), the trajectory, velocity, acceleration, etc., of the centroid at different times in the take-off stage can be calculated, and also the velocity of the RP can be controlled by the relevant parameters in Equation (3). The flight phase of the centroid is calculated according to the conservation of momentum for a body falling freely, and the centroid trajectory of the RP is a periodic trajectory comprising the take-off and flight stages (Figure 6c).

The joint settings of the humanoid robot are highly similar to the TNO HBM. We applied the running process equation of the humanoid robot to the TNO HBM to realize the running process of pedestrians (Figure 6b). In designing the trajectories of the pedestrian’s arm and lower body, the assumption is that they do not interfere with the body. The method of cubic spline curves and the centroid trajectory of the pedestrian are combined into a set of nonlinear equations [31,32,33], and the position, velocity, and acceleration of each joint at every moment are obtained by solving these equations, thereby realizing the running of the pedestrian, which ignores the degree of freedom of the elbow joint with the arm.

Using the rigid–flexible coupled digital HBM, the unknown kinematic parameters are obtained using the multi-body dynamics software MADYMO according to the humanoid robot method and then incorporated into the HBM to simulate and capture 10 postures of the running model (Figure 7).

## 3. Examples and Results

A car–running pedestrian TA was simulated, and the initial state of the RP simulation was set using different methods. The effect of changing the initial conditions on the accident reconstruction results was evaluated, and the RP based on the rigid–flexible coupled HBM was verified in reconstructing the TA. The initial motion states of three typical pedestrian simulations were analyzed, as given in Table 1.

The reconstruction of a pedestrian–vehicle collision accident based on the running state was divided into two stages: First, the motion characteristics of the initial parameters in the pedestrian collision were determined, and the RP dynamic process was pre-simulated to obtain the motion states of the joints for various parts of the human body at the moment of collision. Second, the first-stage simulation results were input into the pedestrian–vehicle collision simulation model as the initial conditions, and the accident was reconstructed.

### 3.1. Accident Case Information

The TA occurred on a pedestrian zebra crossing (Figure 8a). A black car was travelling normally from north (the top of the figure) to south (the bottom of the figure); as it was about to drive over the zebra crossing, it collided with a pedestrian who was running quickly from west (left) to east (right). The pedestrian had not seen the approaching vehicle and was trying to cross the zebra crossing quickly in case of a red traffic light on the sidewalk. A row of cars that had stopped by the side of the road was blocking the line of sight for the driver of the approaching vehicle; thus, the driver did not see the pedestrian running from the side and did not apply the brakes in time; both were travelling relatively fast, so there was a collision. After the collision, the car continued to move forward and turned left for a distance, and then stopped. The pedestrian was hit and flew up and landed on the road, stopped after sliding a certain distance on the ground, and died on the spot.

The pedestrian in the accident was a 25–30-year-old man who was 173–175 cm in height and weighed about 68 kg. An autopsy found that the pedestrian’s lower left limb and knee bone were fractured, their back had suffered serious abrasion, their left elbow and arm had suffered slight abrasions, and their head had suffered a craniocerebral fracture and brain tissue contusion due to the collision.

### 3.2. Determination of Initial Accident Parameters

The initial parameters of the car–running pedestrian collision simulation in the running state include the collision velocity, pedestrian posture, collision angle, and so on at the moment of collision [15].

#### 3.2.1. Initial Velocity of Accident Vehicle

Based on the China’s national standards for zebra crossings, PhotoModeler was used to project pictures captured by the accident video to obtain the distance moved by the vehicle over a period of time. The initial velocity of the car was then calculated according to the time interval of the video [5].

Two pictures were captured from the video at frames 218 and 227 (Figure 8a,b). The PhotoModeler measurements showed that the distance travelled by the car between those two frames was 6.628 m, and the frame rate of the video was 25 fps; therefore, it can be concluded that the car was travelling at 18.41 m/s at the time of the collision. Similarly, the pedestrian was running at 4.168 m/s. However, the clarity of the video and other factors may have affected the results, so a deviation of ±10% is possible [26].

#### 3.2.2. Pedestrian Collision Posture

We extracted a frame from the video taken before the collision and used PhotoModeler software (https://www.photomodeler.com/) to measure the vehicle and pedestrian position information. According to the position information from the video, the velocities were applied to the car FM and the rigid−flexible coupled HBM in MADYMO software, and the FE−MB rigid−flexible contact characteristics and the contact characteristic curves of the car were established [34]. As given in Table 2, the simulation results provide the angles and angular velocities, respectively, of the main joints at the moment of pedestrian collision, where R1, R2, and R3 are the rotation angles of the joints around the ξ, η, and ζ axes, respectively [35].

The running posture at the critical moment of the collision was compared with the video pictures in Figure 9a,b. The simulation results are almost consistent with the real results of the accident, which further verifies the feasibility of the pedestrian model and prepares for the next phase of the accident reconstruction model.

### 3.3. Accident Case Simulation

The TA model in this simulation is a pedestrian–vehicle–road coupled model including the road surface and zebra crossing. The simulation results from the first stage were input into the car–running pedestrian collision simulation model (Figure 9c–e). Then, the velocities are applied to the car and the pedestrian, consecutively, and the friction coefficients between the car and the ground, the pedestrian and the ground, and the car and the pedestrian were defined as 0.7, 0.6, and 0.3, respectively [4,5]. Finally, after several simulation iterations, the information about the pedestrian motion posture and rest position under different vehicle velocities, running velocities, and collision angles were obtained. Compared to the video of the collision, the following conclusions can be drawn: when the angle between the vehicle and the sidewalk is 2.7°, the car velocity is 18.5 m/s, and the pedestrian running velocity is 4.5 m/s. The pedestrian motion posture and rest position are basically consistent with the video, which proves that the physical process of the collision between the pedestrian and the car can be reconstructed accurately in this case.

The simulation results show that the pedestrian’s left leg collided with the car’s right lamp in the middle of the zebra crossing, and then the pedestrian’s upper back collided with the car’s windshield. After that, the pedestrian was rotated forward in the air to a height of 4.2 m and then fell to the ground; finally, the pedestrian stopped after sliding for 5.38 m on the ground.

### 3.4. Human Injury Analysis

In a car–running pedestrian TA, the pedestrian’s head and lower limbs are the most vulnerable, and craniocerebral injury is the most critical factor in disability and even death of the pedestrian [36]. The important criteria for judging the craniocerebral injury and lower extremity fractures are (i) the head injury criterion (HIC) [4] value and (ii) the von Mises stress of the lower extremities [37]. The simulated injury results were compared with the forensic autopsy results as shown in Table 3.

The acceleration curve (HIC = 3913) for the pedestrian’s head far exceeds the HIC limit that the human body can bear (Figure 10a). This occurred in the first collision with the car windshield and the second collision with the ground, each of which was enough to cause severe craniocerebral injury. The simulation results are in line with the forensic autopsy results (Table 3). The maximum value of the von Mises stress distribution nephogram at the moment of collision with the pedestrian’s left leg is 389.5 Mpa (Figure 10b), the maximum stress at the tibia and femur of the pedestrian’s left lower limb exceeds the fracture tolerance limit of 124 MPa [37], and the maximum stress at the fibula is close to the tolerance limit. This is similar to the autopsy result, in which both the tibia and femur of the left lower limb were fractured (Table 3). The reliability of the accident simulation results is further shown.

### 3.5. Case Comparative Analysis

#### 3.5.1. Impact Analysis of Thrown Distance and Rest Position

The initial parameters of the accident simulation were obtained in the preceding part of the paper, and the accuracy and reliability of case 1 have been verified. We used the same method to simulate cases 2 and 3 (Figure 11a). In case 1, the joint angles and angular velocities of the RP were applied (Table 2), and the initial velocity was 4.5 m/s. In case 2, only the initial velocity is applied, and the posture is static (no joint angles). In case 3, the RP posture is static. The pedestrian rest position (Figure 11b), finial position (Figure 11c), and thrown distance were compared among the three simulations.

#### 3.5.2. Impact Analysis of Pedestrian Injury

The primary concern regarding pedestrian injuries is about body parts in the collision who injury causes death and serious injuries, including the head, chest, and lower limbs. Pedestrian head injuries is evaluated using the HIC value, which can be obtained from the head acceleration curve (Figure 12a). Thoracic injury is usually evaluated using the 3 ms acceleration curve criterion (Figure 12b). Lower-limb injury is evaluated using the equivalent stress nephogram of the lower-limb FE method (Figure 12c), and the maximum injury value and the moment of occurrence can be described using the lower limb torque curve, the shear force curve, and the axial force curve. Figure 12d and Table 4 show a comparative analysis of the relative error about the maximum injury values in the three cases. 

## 4. Discussion

To simulate car–running pedestrian TAs, studying the motion state of the pedestrian and the running process before the collision will greatly increase the difficulty of the simulation. In this study, based on the humanoid robot method, we reconstructed the pedestrian running process and applied it to a traffic case. We validated the feasibility of this method by comparing injuries with the forensic autopsy results and by comparing pedestrian dynamics with CCTV footage. In addition, we compared a case without considering the initial state and analyzed the relative error.

Comparing the rest positions (Figure 11b) of the three cases, we found that there is little difference in the rest position of the pedestrian in cases 1 and 2, but there is an obvious difference between case 3 and the other two cases. Therefore, we can deduce that the pedestrian’s velocity is likely to affect how the body is thrown obliquely in the direction of running and affect the angle at which the body is thrown. This is in line with previous studies [13,38]. The pedestrian’s final position in case 2 is 2.13 m, different to that of the other two schemes, and the angle difference is 9.5° (Figure 11c). Hence, we predicted that the RP’s collision posture and joint angular velocity are likely to affect the final position and landing posture of the human body. This is similar to the previous studies which found that gait is likely to affect the trajectory of pedestrians [23] or humanoid robots [9].

Comparing the head, chest, and lower limb injuries in the three cases (Figure 12 and Table 4), we found that case 1 is obviously higher than the other two cases for the maximum head and chest injuries, but the fluctuation trends in the injury values are approximately the same (Figure 12a,b). This is in line with a previous study [38] in which the pedestrian’s initial posture and velocity affected the head acceleration value. In addition, the lower limb fracture injury value of case 2 is obviously higher than cases 1 and 3, but the stress distributions of the three cases are basically the same (Figure 12c). This is in good agreement with studies [16,24], showing that a pedestrian’s posture and velocity can affect the stress and strain on knee ligaments. However, if we focus on the impact of the initial motion state on pedestrian injuries, more simulation and experimental research is needed. We simply made a comparison with the usual two cases.

Although the RP’s movement posture and injury after collision were reconstructed in the TA case, the current research still has some limitations. The first that should be considered is that the accident reconstruction must be completed in two processes due to the limitation of the software’s algorithm function. This may complicate the accident reconstruction process and make it cumbersome. Another limitation is that we can accurately apply the velocity, but it is difficult to ensure the trajectories of the arms and limbs in the HBM are consistent with the RP. The movement trajectories of pedestrian limbs are often different at different velocities; thus, we did not take this aspect into account when validating the RP. The third is that the authors did not consider the limitation originating from biological variability. Although the rigid–flexible coupled HBM has a high fidelity and is scaled according to the victim’s height and weight, there may be biological differences between different individuals, and it is difficult to ensure that the biomechanics of the HBM are completely consistent with the victim. In addition, this paper does not consider pre-collision reactions such as compensation behavior and the reaction times of drivers and pedestrians. These factors will have a certain impact on the outcome of the accident. It is worthwhile examining all these issues through further studies.

## 5. Conclusions

In summary, this study proposed a novel approach for reconstructing moving pedestrian TAs based on a humanoid robot method. To improve the accuracy of car–running pedestrian accident reconstruction, we applied the theory of humanoid robots to the corresponding joints and centroids of the TNO HBM and simulated the pedestrian running process. We validated this model by reconstructing an RP case. The comparison of the simulated dynamics and pedestrians and the comparison of simulated injury and forensic autopsy results both validated the feasibility of the reconstruction method of RPs. In contrast to two other cases, there are relative errors in the pedestrian injury and the rest position. This comparative analysis is helpful for understanding the differences in injury characteristics between the running pedestrian and the other two cases in TAs.

## Figures and Tables

**Figure 1 sensors-23-07882-f001:**
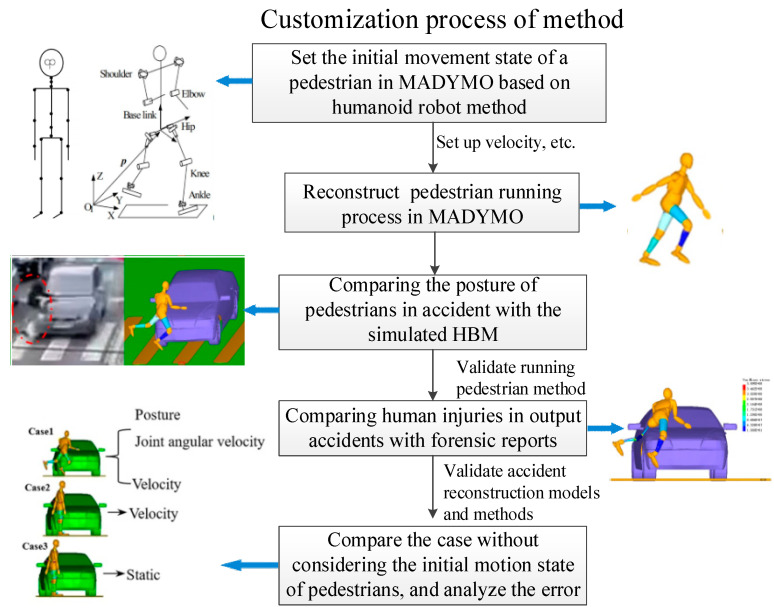
The equivalent stiffness curve of the vehicle surface.

**Figure 2 sensors-23-07882-f002:**
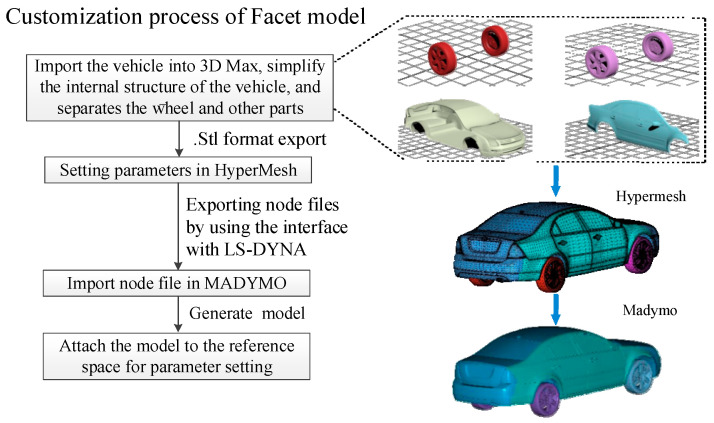
Facet model of the accident vehicle.

**Figure 3 sensors-23-07882-f003:**
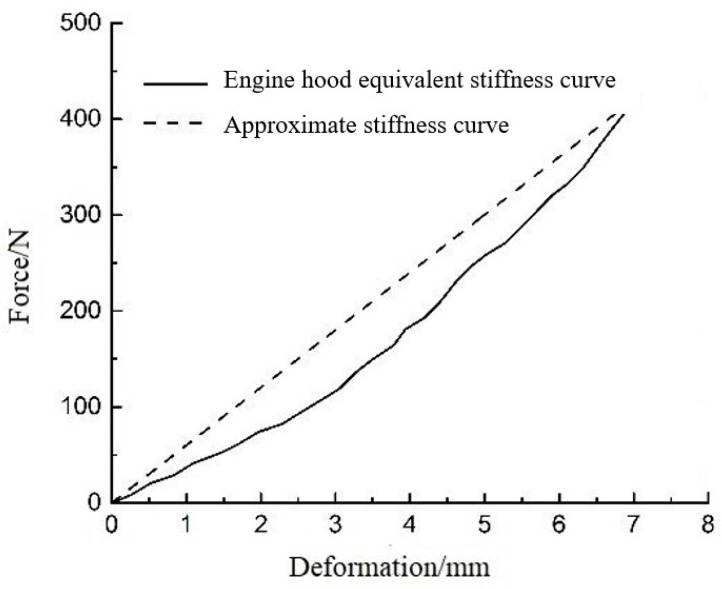
Curves of contact characteristics for the vehicle and pedestrian.

**Figure 4 sensors-23-07882-f004:**
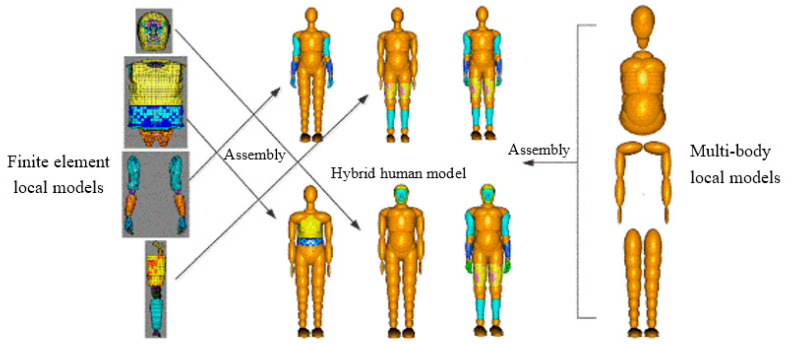
Customization of the mixed structure of the rigid–flexible coupled modular HBM.

**Figure 5 sensors-23-07882-f005:**
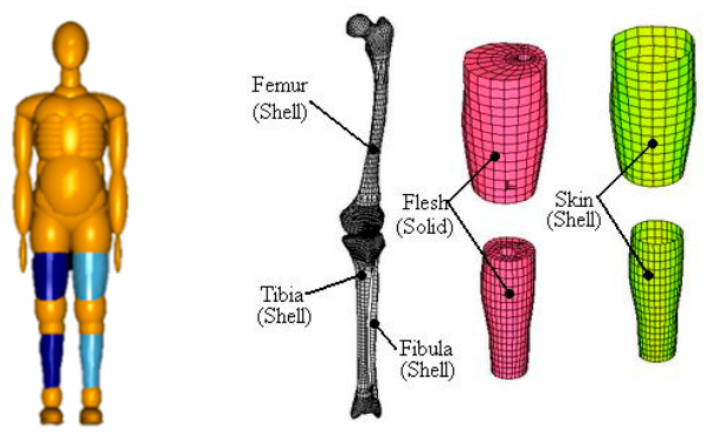
Rigid–flexible coupled HBM (**left**) and FE model of leg (**right**).

**Figure 6 sensors-23-07882-f006:**
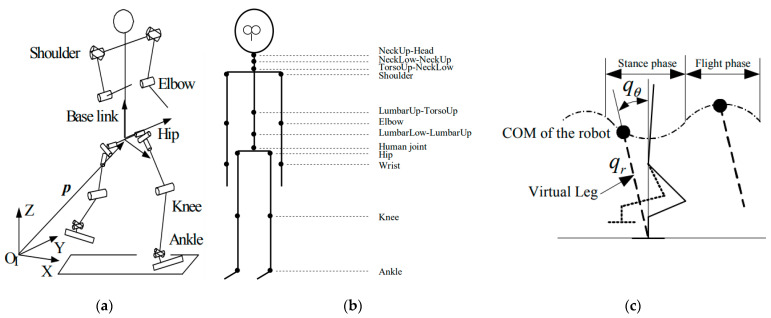
(**a**) Joint structure of robot. (**b**) Joint structure of the TNO HBM. (**c**) Virtual leg of the robot.

**Figure 7 sensors-23-07882-f007:**
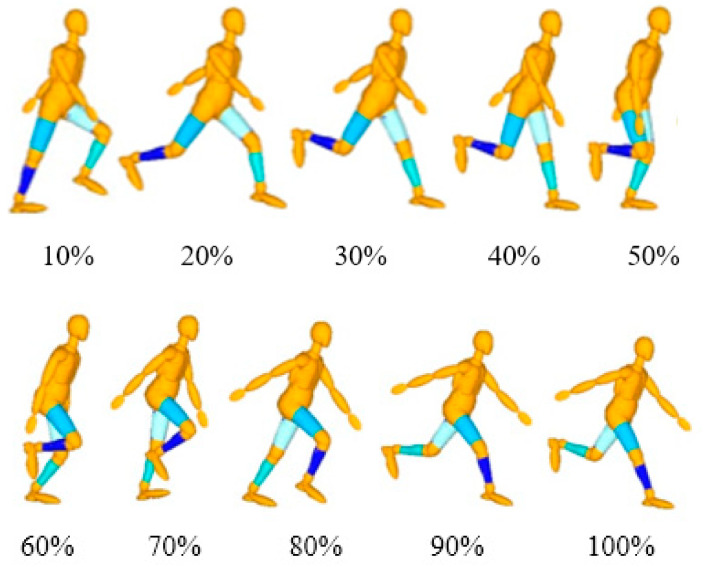
Simulation of running pedestrian.

**Figure 8 sensors-23-07882-f008:**
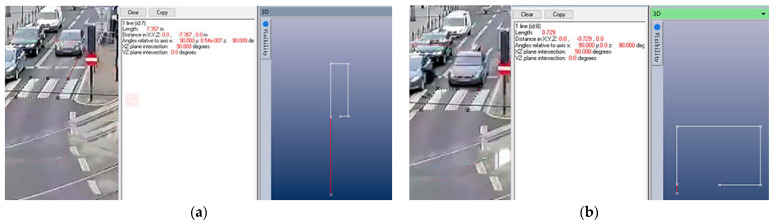
Position of car in frame (**a**) 218 and (**b**) 227.

**Figure 9 sensors-23-07882-f009:**
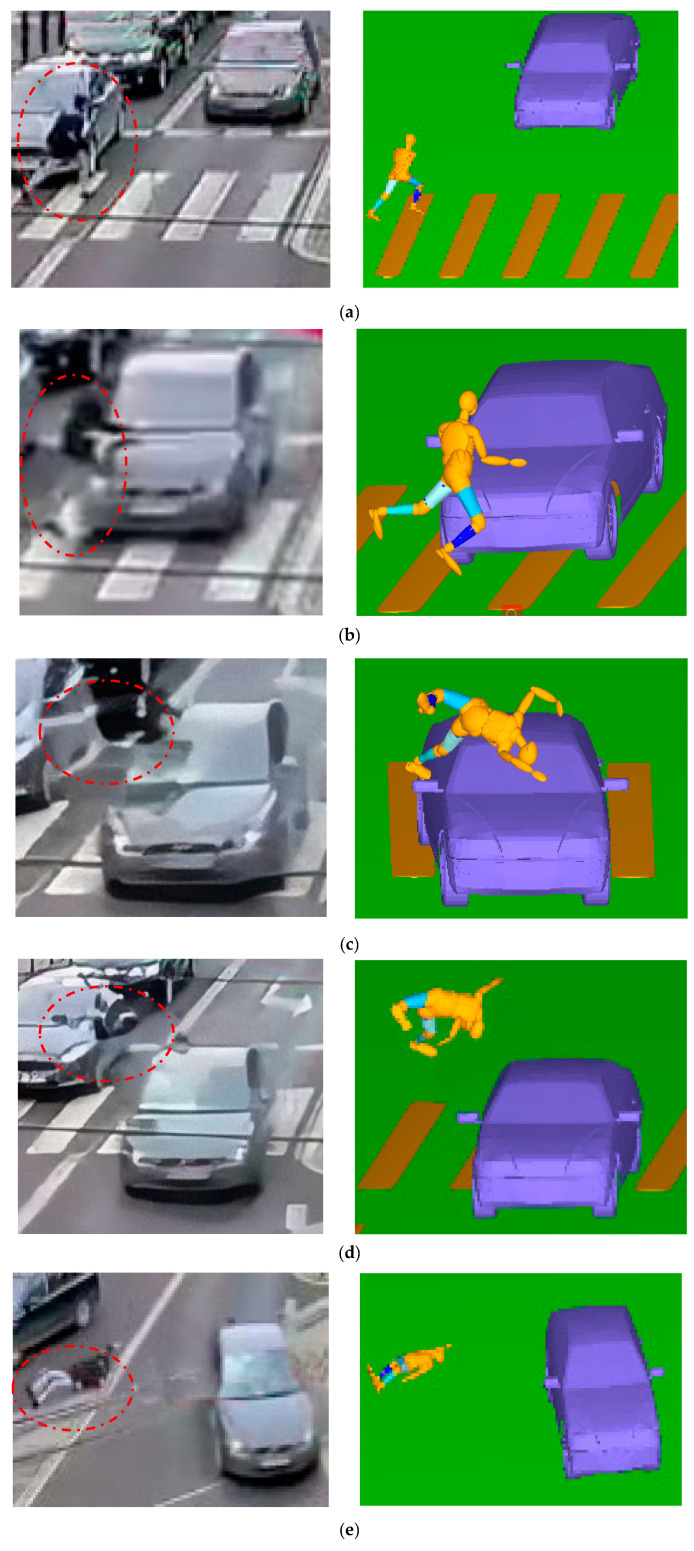
Comparison of critical moment simulations and actual results: t = (**a**) 500 and (**b**) 850 ms. Car–running pedestrian collision physical process: t = (**c**) 140, (**d**) 200, and (**e**) 1500 ms.

**Figure 10 sensors-23-07882-f010:**
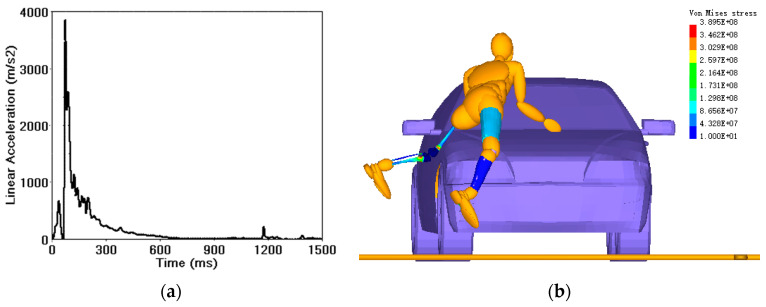
(**a**) Head acceleration curve of pedestrian. (**b**) von Mises stress distribution of lower extremity.

**Figure 11 sensors-23-07882-f011:**
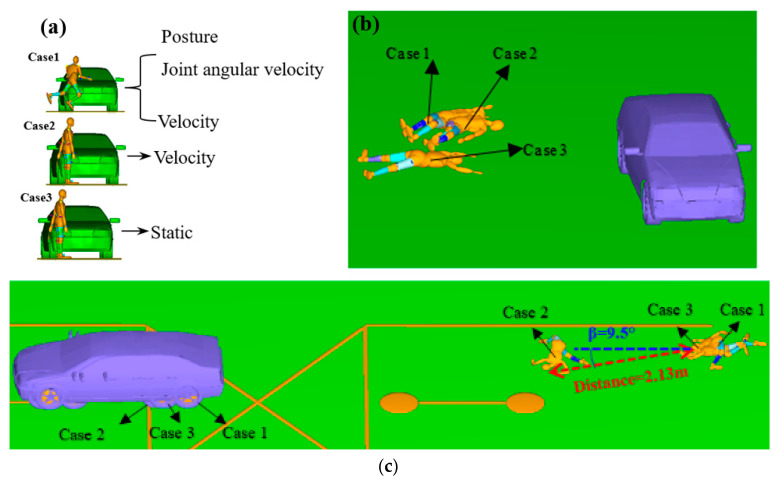
Comparison of pedestrian: (**a**) initial collision conditions, (**b**) rest positions, (**c**) final positions.

**Figure 12 sensors-23-07882-f012:**
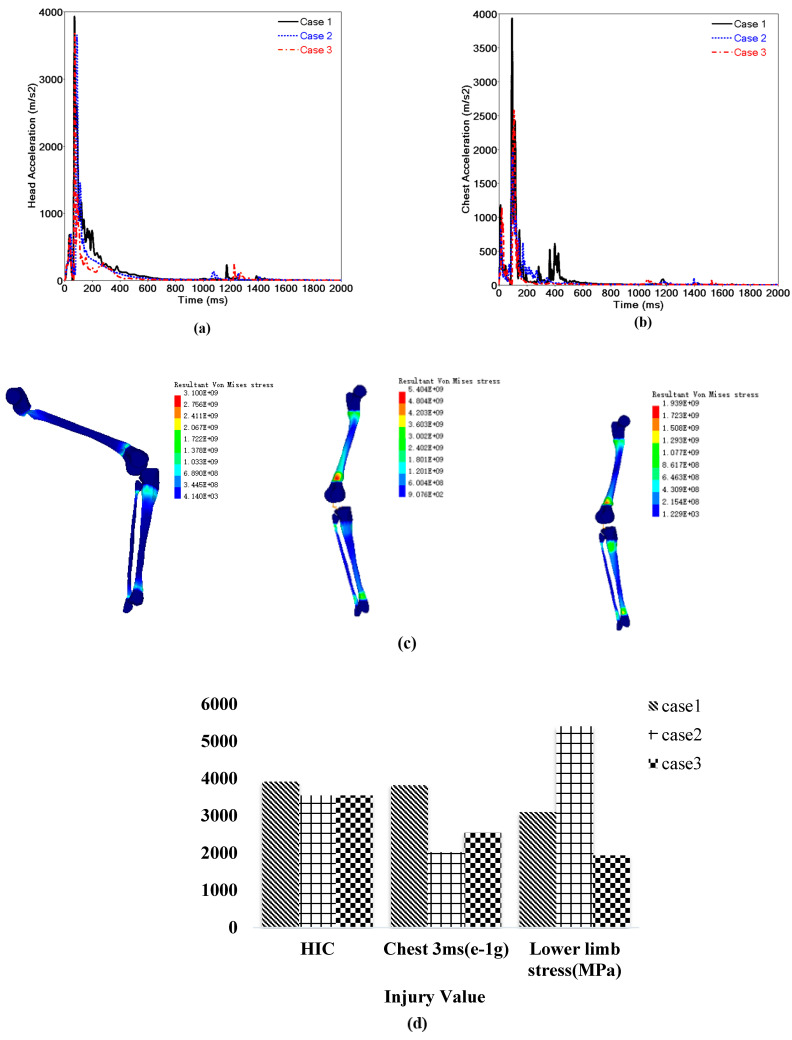
Comparison of (**a**) head acceleration curves, (**b**) chest acceleration curves, (**c**) lower limb von Mises stresses, (**d**) human injury values.

**Table 1 sensors-23-07882-t001:** States of initial pedestrian motion.

Case	Pedestrian Running Posture	Pedestrian Velocity
1	Consistent with running dynamic process	Consistent with running dynamic process
2	Known accident information statistics	Overall translation initial velocity
3	Known accident information statistics	Static condition

**Table 2 sensors-23-07882-t002:** Angles and angular velocities at time of collision.

Body Joint	Angle(s) (rad)	Body Part	Angular Velocity (rad/s)
HipL	*R*_2_ = 0.875	Lower left arm	2.0305
KneeL	*R*_1_ = 0.56	Upper left arm	2.2748
TibiaL	*R*_1_ = −0.015; *R*_2_ = −0.205	Lower right arm	2.0295
HipR	*R*_2_ = −0.345	Upper right arm	2.2739
KneeR	*R*_1_ = 1.37	Left thigh	2.0335
ElbowL	*R*_2_ = −1.045	Left calf	4.7907
ElbowR	*R*_1_ = 0.52; *R*_2_ = −0.965	Right thigh	3.0025
LumbarLow-LumbarUp	*R*_1_ = −0.785; *R*_2_ = 0.305	Right calf	1.2093
NeckLow-NeckUp	*R*_2_ = −0.25	-	-
ShoulderL	*R*_1_ = −1.61; R_2_ = 0.15708	-	-
ShoulderR	*R*_1_ = −1.645; *R*_2_ = −0.232	-	-

**Table 3 sensors-23-07882-t003:** Comparison of simulation results and forensic autopsy results.

Body Part	Autopsy Results	Injury Value of Simulation Results	Injury Limit Value
Head	Depressed fracture of skull, contusion of brain tissue	HIC = 3913	1000
Left lower limb	Fractures of tibia and femur of left lower limb	Maximum shear force: 389.5 MPa	124 MPa

**Table 4 sensors-23-07882-t004:** Comparison the relative errors of three cases.

Relative Errors	HIC	Chest 3 ms	Lower Limb Stress
Case 2 with case 1	9.4%	47.0%	74.3%
Case 3 with case 1	9.4%	33.4%	37.5%

## Data Availability

Supporting documents for this article can be found at the link: https://zenodo.org/record/8162467, accessed on 11 September 2023.

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
