# Peer review of "Reconstruction of a Car–Running Pedestrian Accident Based on a Humanoid Robot Method"

_sensors, 2023, doi:10.3390/s23187882_

Round 1
Reviewer 1 Report
The materials and methods section of the article is hard to follow and understand. The main issue is that in the present form, it is unclear what the author's mean by the term "humanoid robot method", why it was used, and what its advantage is.
The abbreviation "TNO" is used multiple times, but its meaning is missing from the manuscript.
The authors use the term "forensic identification" incorrectly. I suggest using the term "forensic autopsy" instead of it.
The authors describe some limitations in the discussion, but they don't mention the main limitation originating from biological variability:
- It is possible to make a model of a running pedestrian, but real life usually differs from that model
- It is possible to make a biomechanical model of an average human, but it is impossible to make a model of a given human (the victim involved): body proportions and gravity center differ in different individuals, and there are also differences in joint movement limits; in and even the actual muscle stiffness determines the movements of the bodyparts
- the same thing applied to the injuries: there are great individual differences in the strenght of tissues.
There is also some strange issue with the formatting: line numbering starts at page 6. Later, the line numbering just repeats itself multiple times.
Table 4 has to be adjusted to the left.
The language is generally good, but some sentences are very hard to understand.
Author Response
Thank this Reviewer for this constructive comment.

Reviewer 2 Report
This paper presents a novel approach for reconstructing motion pedestrian traffic accidents based on humanoid robot method to improve the accuracy of analyzing dynamic vehicle-pedestrian collision accidents. Overall, the paper is interesting and sound. The work is organized well and the logic looks nice. However, there are some points that should be addressed to improve its presentation and scientific content:
1. In section Introduction, if the structure of the article is introduced in the last paragraph, it will be more helpful for the reader to understand.
2. Figure 3 should be optimized. Note is not fully displayed.
3. The three-line table in the main text is incorrect. The format in the appendix is correct.
4. Line 51 on page 2 should be expressed as “Two pictures were captured from the video at frames 218 and 227 (Figure8 (a) and (b))”.
5. There is a formatting error on page 5, line 146.
6. Try to avoid expressions such as "Figure 10 (a) shows xxx" on page 5, line 113, and page 7, line 175.
Minor editing of English language required
Author Response

(The authors gave the same response as above.)

Reviewer 3 Report
The authors of the manuscript presented their original new research achievement, based on the use of the principle of running hu-manoid robot to the digital human body model to realize the running and walking process of pedestrians, in a car–running pedestrian accident.
Comments:
1. In the Introduction, after the well-formulated purpose of the article, there is no indication of who can benefit from these research results.
2. In Section 2.3.3, in the description of formula (3) it reads ".... dynamic equations of the motion trajectory ...", and it should be "... differential dynamic equations of the virtual leg motion whose solution is its motion trajectory ...".
3. In Figure 12d, enter the units of human injury values.
4. Introduce Section 5. Conclusions, in which to collect the most important conclusions from the research, and above all, indicate a detailed plan for further research in the subject of the paper.
5. Five pages of the manuscript are missing!?
Author Response

(The authors gave the same response as above.)

Round 2
Reviewer 1 Report
Point 1:
I accept the answer about the definition of human robot. The added sentences solve the issue.
Points 2:
All abbreviation (as TNO) should be explained not by a citation but with writing it’s meaning fully when it first mentioned in the abstract and manuscript - like „finite element (FE)” in the first line of the manuscript. Please add it the full name of TNO to line 17 and 108.
Points 3 to 6:
These points were adequately addressed.